# The Value of Public Sector Risk Management: An Empirical Assessment of Ghana

**Yusheng Kong [1], Peter Yao Lartey [1,\*], Fatoumata Binta Maci Bah [2] and Nirmalya B. Biswas [3]**

[1]    School of Finance & Economics, Jiangsu University, Zhenjiang 212013, China; yshkong@ujs.edu.cn
[2]    School of Biological Engineering and Life Sciences, Jiangsu University, Zhenjiang 212013, China; mariamatimbi@yahoo.com
[3]    London School of Commerce & ISBC, Bangalore 560037, India; nirmalya.b@isbc.ac.in
\*    Correspondence: efopeter@yahoo.com; Tel.: +86-156-951-17992

**Abstract:** This study investigates risk management practices in public entities in the Ghana. We relied on the popular framework designed by the Committee of Sponsoring Organizations of the Treadway Commission—COSO, to advocate for possible ways to minimize the occurrence and effects of risk in public organizations. The internal control elements used include: control environment, commitment to ethics, segregation of duties, review and information and communication. These constitute the explanatory variables used in performing multivariate data analysis to determine the dimensionality of the data set and possible outcomes. The exploratory research followed a quantitative approach using the survey method and a structured equation model. We established that, due to globalization and increases in the scale of operations, it is practically impossible for management through the help of auditors and those in charge of governance to validate the entire operations of the public sector to ensure strict compliance to internal control principles, in order to minimize the detrimental impacts of risk. However, an alternative sustainability depends on the prominence of quality financial reporting, compliance, commitment to ethical values and consistency in pursuit of the strategic and operational objectives based on good corporate governance. On the other hand, the implications of risks should be embedded in the minds of public servants as part of the organizational culture that will complement existing tools and techniques of internal control.

**Keywords:** public sector; Committee of Sponsoring Organizations of the Treadway Commission (COSO–Enterprise Risk Management); risks assessment; control environment

## 1. Introduction

The emergence of organizational risks management has gained significant popularity in theory and real life application in recent years regarding the performance of corporate and public sector entities (Ahsan and Rahman 2017; Aliewi et al. 2017; Arena et al. 2017; Ge et al.). A couple of decades ago, it took very large corporate scandals before the concept of internal control, corporate governance and risk management were revisited with positive interventions such as the passing of Sarbanes Oxley Act in 2003 in America, the corporate governance principles developed by the Organization for Economic Cooperation and Development (OECD), the UK Corporate governance code and more recently the updated internal control and risk management framework of the Committee of Sponsoring Organizations of the Treadway Commission—COSO in 1992–2013 (Collier 2009a, 2009b; Hagigi and Sivakumar 2009; Martin et al. 2014). Owing to deficiencies in internal control and the emergence of risk, the corporate environment experienced the most turbulent events involving the failure of Enron, a leading energy company, WorldCom and more recently Lehman brothers, one of the largest banks in the US (Ma and Ma 2011; Daniela and Attila 2013; Arena et al. 2017; Callahan

and Soileau 2017; Da Silva Etges et al. 2018). The main causes of these corporate demises were attributed to non-compliance and disregard for ethical practices, managerial greed, internal control lapse, and weak corporate governance standards (Aziz et al. 2015; Themsen and Skærbæk 2018). There are, nonetheless, distinctive circumstances that warranted the popularity of the concept and its relevance in public sector today as the basis for measuring performance, transparency and value for money (Hey 2017; Demek et al. 2018). Forecasting risks is a necessity for sustainable business and arguably a prime tool and technique for risk assessment, and there is a determination to reduce risks and raise awareness about its occurrence by ensuring compliance to operational standards and initiating conversional practices that will identify and keep risk within acceptable perimeters (Adhikari and Gårseth-Nesbakk 2016; Agyei-Mensah and Kwame 2016; Ahluwalia et al. 2016; Tan 2016). This study is of the view that internal control and for that matter risk management play critical roles in the performance of an organization. However, while internal control is highly popular among public and private entities, the importance of risk management is overlooked in the public sector (Axelsen et al. 2017; Barafort et al. 2017). Therefore, the aim of this study is to investigate the organizational risk model by carefully testing some selected internal control variables that will explain the process of planning, directing and controlling the activities from top to bottom in order to minimize the effects of risk on capital and resources in the public sector (Aziz et al. 2015; Ashraf and Uddin 2016). The study anticipates a big gap between performance and the understanding of risk in the public sector. In order to fill this gap, we ask the research question; "what role does internal control play in addressing organizational risk?" In an attempt to interrogate this discrepancy, the study relied on the COSO internal control framework to develop a conceptual framework which identified five components of internal control that could be used to address risk: (i) commitment to ethical values; (ii) control environment; (iii) segregation of duties; (iv) review; and (v) information and communication. With these elements, we seek to contribute to existing knowledge in theory and practice about the need to reinforce organizational risk management in order to reduce the detrimental effects of risks in public sector administration. This study draws motivation from the findings of the Auditor General of the republic of Ghana, in what is termed the abysmal approach to risk management in the handling of public accounts (Akotia 2016; Asante 2016). It is evident that internal control is fully established in the public sector, however there are challenges with compliance and its linkage to risk management (Abbott et al. 2016; Adhikari and Gårseth-Nesbakk 2016; Arena et al. 2017). The public sector overreliance on external evaluators and the consultancy services of independent third parties would only amount to a short-term solution to the long-term predicament (Lim et al. 2017). If risk management forms an integral part of operations, there is the likelihood that the occurrence of unforeseen events that is contributing to inefficiencies, financial irregularities and misappropriations and waste of resources in the public sector would be kept under control (Barafort et al. 2017). As part of the challenges, the reliability and independence of internal auditors and external auditors who are referred to as the custodians of internal control at various departments, local government divisions and ministries could not be established, looking at the reoccurrence of financial irregularities, misappropriations and waste of public resources as clearly stated in (Asiedu and Deffor 2017; Callahan and Soileau 2017). This paper is organized as follows: Section 1, introduction to the study and prior studies; Section 2 outlines relevant literature and theoretical concepts. Section 3 briefly outlines methodology and data reliability. Section 4 contains the results, analysis of findings, discussion and implication. Section 5 presents the overall conclusion.

*Prior Studies*

A research conducted by Scheers et al. (2006), forms part of the historical studies on international internal control systems in the public sector. The study contributed to existing findings on the application of internal control frameworks in different countries believed to be the underpinning theories at the time. Following this development, Australia established what is called the central control model based on the five traditional elements of the COSO internal control framework, to

guide its public sector against risks and possible lapses in administration (Ahluwalia et al. 2016; Akotia 2016; Maudos 2017). In Sweden, public sector administration is based on a mixed method (Scheers et al. 2006). It relies on the COSO internal control framework, the opinions of policy makers and the central government's own specific procedures based on the size, work, governance structure and organizational internal factors (Domingues et al. 2017). This is in sharp contrast with the case of Ghana, where the public sector is strictly governed based on laws, administrative and legal instruments that are subject to amendment only through parliamentary procedures, making it extremely cumbersome to implement innovation flexibly in this fast-changing globalized world (Appiah et al. 2016; Asante 2016; Barafort et al. 2017). Whereas in the US, enterprise risk management in public sector administration is mostly guided by the COSO frameworks for risk and internal control systems, proposed by five private sector organizations committed to providing guidance through the development of frameworks to support corporate business and the public sector (Adhikari and Gårseth-Nesbakk 2016; King 2016; Basu 2016). The International Organization of Supreme Audit Institutions (INTOSAI), which functions as an umbrella organization for the entire government audit community, and a couple audit firms such as KPMG, Deloitte, Ernest & Young and PricewaterhouseCoopers have contributed immensely to risk management as far as internal control and public sector administration is concerned (Lim et al. 2017). As a result, the internal control systems within the governance structure of the United States, provides a holistic direction to performance measurement and the identification of major risks and challenges associated with public sector performance (Scheers et al. 2006). In principle, these standards of internal control systems play key roles in organizational risk management, prevention and detection.

## 2. The Concept of Risk

Every business faces risks, especially those operating at high capacity with a vast number of employees and resources like the public sector. That is why every organization implements risk assessment to identify threats and then manage them efficiently since business conditions are constantly changing (Basu 2016; Baumgartner and Rauter 2017; Da Silva Etges et al. 2018). It is relevant to perform risk assessment to identify the internal and external threats that the entity is exposed to (Dementiev 2016; Dong et al. 2017). The risks could appear in different forms; for example, an entity can be confronted with financial reporting errors, fraud, irregularities or inadequate workforce training; however, external risks could refer to changing and complex consumer demands, new competitors, or possible natural disasters (Balabonienė and Večerskienė 2015). Assessing risks includes estimating the impact and the ability to accurately estimate the likelihood of its occurrence and then assessing appropriate actions to reduce its impacts. This is done bearing in mind the organization's readiness in terms of resources, personnel and finance to counter risks (Bromiley et al. 2015). Evidence from research indicates that enterprise sustainability is largely dependent on the ability of management to envisage and forecast risk to avoid potential occurrences (Johansson and Siverbo 2014). Risk management is a deliberate process involving the identification, assessment and prioritizing of risks, taking into account both positive and negative implications followed by a commitment of resources to mitigate and monitor in order to keep the risks under control to counter any unfortunate happenings with the intention of maximizing rewards (Lerskullawat 2017). However, due to the challenges associated with the process, which mainly are to do with a scarcity of resources, most firms prepare a list of risks with the most pressing ones highly preferred to less important risks in order not to render the procedure unbearable for organizations so as to be strictly practice (Afiah and Azwari 2015; Dong et al. 2017). Since risk management process is so demanding and quite expensive such that an organization is committing more resources into managing risks, the outcome must create value for the public sector rather than contribute to waste of resource at the detriment of the taxpayer (Zins and Weill 2017). Drawing from a school of thought in management literature, risk management principles should be an integral part of the organizational processes in order to achieve success, more so as the organizations grows from stage to stage and the types of risks also change, meaning corporate governance strategies

must be aligned to approve change in order to deal with the dynamic nature of risks in a more advance and innovative approach (Apergis et al. 2016).

## 2.1. Events Identification

It is mandatory for an organization to envisage all manner of events emanating from the external environment or internal sources according to Sprčić et al. (2015). Bromiley et al. (2015), investigated the relationship between control activities and risks control in an empirical study, and concluded that unexpected events or occurrences are capable of truncating the realization of the entities objectives if monitoring and reviews are not properly done. Lim et al. (2017), also concluded a similar study indicating that the presence of internal control does not always guarantee absolute assurance of the absence of risks unless policies become an obligation to enforce compliance followed by a periodic validation of the entities operations by auditors (Pellegrini et al. 2017). In a related study, Horne (2017), asserted that overreliance on external evaluators instead of intensifying a robust internal control structure sustained by a good corporate governance principle can spell failure. First of all, the standard procedures applied by the auditors or any third party is largely dependent on the quality of evidence gathered at a particular time. Earley et al. (2016), explained that a sound judgment is based on the auditor's knowledge and experience. By this development, management is encouraged to employ the services of internal auditors and supervisors to ensure that control activities such as segregation of duties and access control is enforced. Ideally, the negative impacts of risks and unforeseen events will be termed as threats while the positives symbolize opportunities that will feed back into strategy (Asante 2016). Meanwhile, management must ensure that operational lapses do not disrupt the entity's strategic objectives. Management should be in the position to identify anything that possesses danger to achieving the entity's strategic objective. The concept is clearly illustrated in Figure 1, where the main component of the COSO framework would help us visualize the whole organization enterprise risk management model and then concentrate on the linkage between internal control and organizational risk management.

**H1.** *Segregation of duties will positively influence risk response as a control activity.*

**H2.** *Consistent review will increase the chances of identifying risks and internal control weaknesses.*

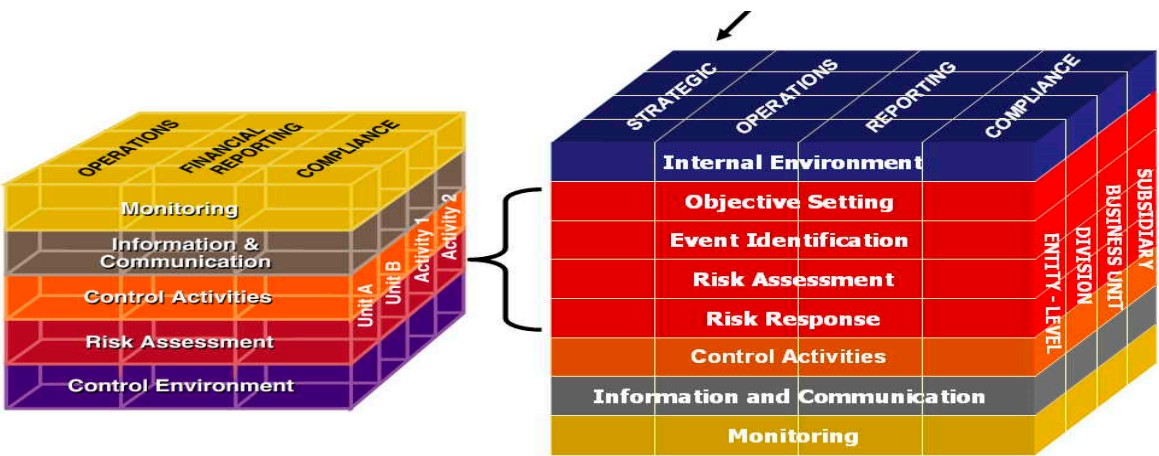

**Figure 1.** Committee of Sponsoring Organizations of the Treadway Commission (COSO) Updated internal control and Enterprise risk management framework, as cited in (Jill and Houmes 2014).

## 2.2. Risk Response

This practice ensures that management selects an appropriate actions in response to managing risks and in alignment with risk tolerance and risk appetite (Ahluwalia et al. 2016). This ensures that management is selecting an appropriate approach to manage risks that is acceptable across all

levels of the entity (Herron and Crawford). The risk response that is chosen must be realistic and take into account the cost of responding as well as the impact on risks and the resources capability of the entity. A study by Asiedu and Deffor (2017) stated that it is important to avoid handling risks in isolation, but rather to consider the entity as a whole. Consequently, risk response will contribute to an effective internal control which could be applied in the form of detective, preventive or automatic controls (Lim et al. 2017). With the precise control environment in place, it could possibly represent the platform on which the other elements of control would function (Blok et al. 2015; Boe and Kvalvika 2015; Bromiley et al. 2015). In responding to risks, it is imperative to observe the control environment as a reflection of the organizational culture built around tolerance and commitment to ethical values and a framework that predicts or lays the foundation for the free flow of business and work (Lerskullawat 2017). In the absence of a control environment, all other elements cease to function as specified by Pellegrini et al. (2017). Therefore, a control environment sets a holistic picture of the entire entity such that management could figure out any strategic change. This action is based on the present system of corporate governance, ethical standards and values pertaining to a particular system of leadership in charge of an entity's finance, economic, social and political affairs (Mensah and Premaratne 2017).

**H3.** *Commitment to ethical values will minimize organisational risk in the public sector.*

**H4.** *Control environment has a positive correlation with risk and events identification.*

### 2.3. Risk Assessment and Internal Control

The International Organization of Supreme Audit (INTOSAI) whose members are the primary external auditors of the United Nations, took a decision to upgrade its earlier rules published in 1992 to guide public sector administration as stated in Barafort et al. (2017). Subsequently, the set of guidelines extensively prescribed preventive and detective measures to minimize the occurrence of risks and possible impacts (Chang et al. 2014). Subsequently, it ratified the guidelines prescribed by COSO, as the internationally recognized internal control and risks management model due to its comprehensive approach to enterprise risk management (Dong et al. 2017). Countless events may account for conditions that could impede an organization from achieving its targets, irrespective of whether private or public, potential risks are capable of disrupting the smooth performance and efficiency of activities (Demek et al. 2018; Guo and Eschenbrenner 2018). The possibility of such occurrences could be predicted whereas the signal and impact could be adequately projected and measured in such a manner that organizations can mitigate its gravity with the support of a fully functional internal control system (Dănescu and Prozan 2015). Gantz and Philpott (2013), established that the art of risk management is largely considered a precondition for corporate governance and business sustainability. It implies that the entity takes keen interest in risk identification, risk definition, risk analysis and appropriate response. However, public sector entities possess distinctive traits that distinguish them from their private counterparts such as the priority areas in risk management, stakeholder groups, profit orientation and the degree of resources including the human capital (Aziz et al. 2015). Risk management, risk analysis and assessment are terms that are frequently used in the public sector and the business domain but the true meaning is often misconstrued in the public sector, which calls for the role of information and communication in conveying clear instructions of processed information across the length and breadth of the organization (Adhikari and Gårseth-Nesbakk 2016). Nonetheless, the real meanings of these concepts are often used interchangeably by means of mixed interpretations (Anna and Nikolay 2015). The globalization of the economy has resulted in the increased in the scale of operations and change in objectives of both public and private entities over the years. This change has, equally, resulted in the complexity of the nature of risks making the public sector more vulnerable to internal control weaknesses (Balabonienė and Večerskienė 2015). For this reason the risk concept applied in the public sector is considered economic risk, which mostly describes negative events (Hillman et al. 2018). In the public sector, risk is considered an occurrence that represents a potential threat to realizing the entity's achievements. A section of management scholars prefer to classify risks

solely according to the COSO internal control guide which describes risks as: certain events, activities or perhaps failure to perform some activities which are bound to occur in the near future, and when they do occur the impacts are grievous, however in many instances risk could amount to an opportunity for the organization (Themsen and Skærbæk 2018). Risks emerged in the public sector the moment individuals began to perceive a high degree of uncertainty with the use of public resources to a certain point, raising doubts about accountability and responsibility (Steinbart et al. 2018). Agyei-Mensah and Kwame (2016), argued that the risks may not be distributed to those individuals privately who may be responsible for them. Hence, the need to advocate prevention of organizational risks in the public sector.

**H5.** *Communication of relevant information about risk will enhance the importance of risk in the public sector.*

### 2.4. Importance of Risk Management in the Public Sector

Considering the devastating impact of risks, COSO has been consistent in advocating for risk awareness by introducing the integrated internal control framework which also provides the guidelines for enterprise risk management (Onyiriuba 2016; Dong et al. 2017). This concept is essential in the public sector because without suitable assessment of enterprise risk there could be a crisis in realizing long-term objectives and even in public confidence in state institutions (Agyei-Mensah and Kwame 2016). Estimated revenue, profit and budgeted productivity may come under severe threat without proper procedures to guard against any deviation from standing orders (Onder et al. 2016). This argument is seconded by Knechel and Salterio (2016, p.13), thus:

> "Poor controls lead to losses, scandals, failures and damage to the reputation of organisations in whatever sector they are from. Where risks are allowed to run wild and new ventures are undertaken without a means of controlling risk, there are likely to be problems."

Following the argument of Knechel and Salterio (2016), it came to light that the consequences of weak control would not only distort performance and profitability targets but dent management reputation in the midst of scandals and should be a matter of concern (Lim et al. 2017). However, internal control still maintains its traditional role of keeping the entire organization under check through the five elements that controls seek to rectify—company policies, performance standards, financial reporting and segregations of functions (Da Silva Etges et al. 2018; Demek et al. 2018). Review, monitoring, and consistent evaluation by auditors will form part of the risk management activities (Jawadi and Louhichi 2017). Risk management plays a vital role in affairs of any organization in modern times, in ascertaining the priority areas when it comes to assessing the opportunity and threats—internal or externally produced (Agyei-Mensah and Kwame 2016; Dong et al. 2017). In corporate governance and strategic planning, risk management ensures each and every unit of the organization is integrated and forms part of the overall decision and policy approach that will place the organization in a control position, such that activities are performed in an orderly manner (Dănescu and Prozan 2015). Ideally, risks form part of the organization's strategic concern, while internal control represents the organization's set of compliance standards as stipulated by the corporate governance mechanisms (Bromiley et al. 2015). Again, enterprise risk management is more concerned about productivity and performance measurement. A critical examination of risk factors in the public sector is an optimistic stride towards efficiency of administration and corporate governance (Joyce 2015). Achieving optimum utilization of human resource capacity and effective public administration is quite challenging in recent times, despite countless efforts by policy makers to ensure modernization and reforms capable of achieving significant value innovation corresponding to the high proportion of the national budget that goes into appropriation of goods and service for the public sector (Apergis et al. 2016).

### 3. Methodology

The survey adopted structured questionnaires, measured on a five-point (5) Likert scale and administered to a sample of 300 participants made up of public servants from the ministries and departments in charge of delivering various government duties. The questionnaire was drafted in six categories, comprising five independent variables and one dependent variables: control environment, segregation of duties, review, information and communication, commitment to ethics, and explaining effective risk management (Niroumand 2017). The independent variables are meant to explain whether or not policy makers can rely on internal control elements to control risks in the public sector (Qiu et al. 2018; Shao et al. 2018). The sample size was considered adequate enough to guarantee a true and accurate inferential judgment on the predictive power of the independent variables (Niroumand 2017).

We considered only the entities located in the central business district of Accra and subsequently measured the responses from strongly disagree (1) to strongly agree (5) (Van Kleef and van Trijp 2018). In section "A", participants check their age, educational levels, years of experience, field of work and respective departments. Data was analyzed using SPSS 13.0 to estimate the degree of total variance, model fitness and the level of association between explanatory variables and the dependent variable (Grady 2018). Structural equation modelling was applied as the main technique for data analysis due to its consistency in predicting causal relationship between constructs (Grover et al. 2018). The multivariate statistical techniques deployed are discriminant analysis and principal component analysis (Niroumand 2017). The choice of these models is based on their supremacy and consistency in social science research (Brzeziński and Bąk 2015). Cronbach's alpha of reliability was conducted to evaluate the model fitness.

### 3.1. The Conceptual Framework and Hypothetical Assumption

The conceptual framework (Figure 2) suggests that effective internal control policy is likely to influence the organization's risk management functions in order to enhance performance. While it is obvious and theoretically verified that the implementation of internal control principles is positively associated with compliance, prior theoretical inquiries proposed that there may be a possible relationship between internal control elements and effective risk management (Arena et al. 2017). Previous studies also suggest that, this relation could only be established if there exist a fully functional and recognized internal control systems within the organization with a practical sense of approaching risks (Barafort et al. 2017; Basu 2016). Due to the nature of the public sector, where the competitive drivers are different from the private sector, it is necessary to exercise caution when addressing issues related to risk management (Afiah and Azwari 2015). The above assertion is fully consistent with the COSO- enterprise risk management model which cautions organizations against handling risk in isolation of internal control (Akotia 2016). Risk management forms an integral part of internal control in addressing organizational deficiencies. However, internal control focuses primarily on compliance while risk management enhances performance (Basu 2016). Based on this knowledge, the conceptual framework (Figure 2), outlined a set of variables developed from a wide range of literature relevant to the study (Cofie 2016; Davis 2016; Dementiev 2016; De Magalhães and Trigo 2017; Dong et al. 2017; Da Silva Etges et al. 2018; Demek et al. 2018). We hypothesized the implementation of a commitment to ethics, control environment, segregation of duties, review and information and communication as agents of internal control which could significantly influence and minimize the impact of risk on the performance of public entities.

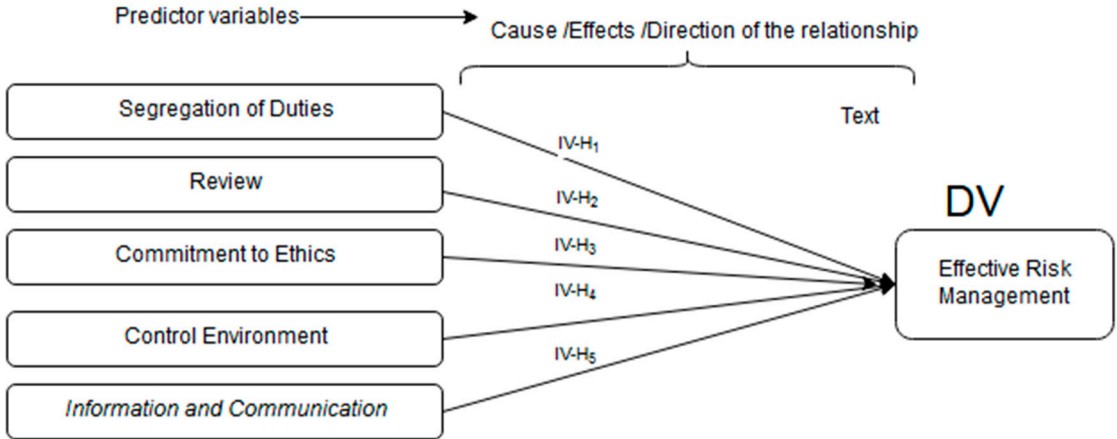

**Figure 2.** Conceptual framework.

*3.2. Reliability and Model Definition*

The study measured reliability of constructs using Cronbach's alpha technique at a margin of 70% threshold (Colbert et al. 2018). Ideally, the acceptable reliability level prescribed by the model is 0.7%. However, the constructs assumed a satisfactory threshold between 0.717 and 0.882 (Table 1), describing the data relatively suitable for further analysis (Menke 2018).

Cronbach's Alpha

$$\alpha = \frac{N \cdot C}{V + (N-1) \cdot C}$$

**Table 1.** Cronbach's alpha of reliability threshold.

| Reliability Statistics | |
|---|---|
| **Cronbach's Alpha** | **N of Items** |
| 0.820 | Commitment to Ethics |
| 0.717 | Control Environment |
| 0.780 | Segregation of duties |
| 0.867 | Review |
| 0.830 | Information & communication |
| 0.882 | Effective Risk Management |

Model Specification

The study relied on the consistency and the robustness of the principal component technique (PCA) and linear discriminant analysis (LDA) in estimating the direction of the constructs in an attempt to report on the degree of varaince and lenear relationship, illustrated in the structural equation as:

$$Y_i = \beta_0 + \beta_1 X_i + \varepsilon_i$$

The prime objective of this study is to appraise the current state of risk management and advocate the awareness of risk to policy makers in the public sector using statistical coefficients as a basis for reliable inference (Shao et al. 2018). Therefore, the impact of the explanatory variables over the dependent variable will be the focus point for our recommendation concerning management's policy to effectively cope with all manner of organisational risks (Qiu et al. 2018).

$Y_i$ = dependent variable–Effectiveness (EFF).

$\beta_0$ represents the population intercept of the dependent variable Y.

$\beta_1$—represents the slope and coefficient of the population ($\beta_1 \ldots \beta_5$).

$X_i$—represents the explanatory variables—(control envin. commitment to ethics, segregation, review and communication)

$\varepsilon_i$ = Random error term

$$EFF_i = \beta_0 + \beta_1 CE_i + \beta_2 CET_i + \beta_3 SEG_i + \beta_4 REV_i + \beta_5 COM_i + \varepsilon_i$$

Since we are not working in the future but relying on the predictive command of the explanatory variables, our statistical inferences will be limited to the efficiency and model fitness to forecast a constant pattern with provision for random error $\varepsilon_i$ (Grover et al. 2018) According to (Colbert et al. 2018), in his original paper where the principal component was first published, he noted that PCA and LDA are two distint techniques with high efficiency rates in social and natural science research.

$$\frac{1}{N} \cdot \sum_{i-1}^{N} Xi = 0$$

where $X_i$ is the vector of one of the $N$ multivariate observations if operated by diangonalizing the coverance matrix (Qiu et al. 2018). Since we attempt to measure the predicting attribute of each construct or group of elements in the case of discriminant analysis, the function will be stated as; c = $\frac{1}{N} \sum_{i-1}^{N} xixiT$. In other words, it produces eigenvalues composition of the coverance matrix represented as: $\tau$ $^{\mathrm{T}}_{\mathrm{T\,V\,=\,CV}}$ which could also stated as; $\lambda_T^T$ V = $X_I^T$ CV□iε(1ᴺ) (Huang et al. 2018). This implies that we will determine the degree of variance and subsequently reduce the dimensionality giving primacy to the model to find the direction of maximum variance, which is consistent with (Grover et al. 2018). Linear discriminant analysis (LDA) equally performs similar functions as a transforming tool; however, the PCA finds the maximum direction variace using the complete data set while the lenear Discriminant anlysis predicts variance accounting for groups of data (Menke 2018). Consider the following $\mathbb{N}\lambda$ = ka where $\mathbb{N}$ is the number of data in the set and $\lambda$ and à are the eigenvalues and eigenfactors of K, in order to standardise the ieigenfactors $a^{k\prime}$s automatically becomes 1 = $(V^K)^T V^k$ this is because ideally the Kermel formular $V_{.}^{kT}$ $_{\phi(X)}$ = $\sum_{i-1}^{n} aik\phi(xi)^T$ $\phi(x)$ of (PCA) is restricted, in that it computes not the principal components themselves but the projections of our data to those components in order to evaluate the projections and subsequent interpretations based of the judgement of the researcher (Basu 2016).

## 4. Result and Implication

### *4.1. Descriptive Statistics*

The study considers organizational risk as an integral part of an internal control function, which is capable of rectifying the recurrent incidents of irregularities and deficiencies at various levels of the public sector structure (Pellegrini et al. 2017). The statistical result will give us a lead to possible ways to address the weaknesses in internal control that can significantly affect risk management. Having this in mind, the study explores the possibility of reliance on the COSO guidelines to measure the weight of each variable to establish a suitable approach for the detection and prevention of risk, as it is done in many developed countries such as UK, US, Europe and some part of Asia and the Gulf countries (Bosten et al. 2017).

#### 4.1.1. Principal Component Analysis (PCA)

Evidence from (Table 2), details the percentage variance by the explanatory variables in an attempt to confirm the importance of risk prevention and detection in the public sector. The outcome is quite significant compared to a previous study at the local government department by Dawuda et al.

(2015), where it was established that internal control variables could not adequately explain auditors' independence of the board in an attempt to assess internal control determinants which is one of the causes of non-compliance. Our study provides statistical evidence that the PCA explained 84.709% variance, attributed to three major constructs. The extracted constructs according to the percentage eigenvalues: $\lambda_T^T$ V = $X_I^T$ CV□iε(1ᴺ), were represented by segregation of duties, commitment to ethics and control environment which are similar to findings in (Hillman et al. 2018; Huang et al. 2018) Meanwhile, the PCA has also produced an interesting outcome by reaffirming the relationship between risk management and internal control as a joint concept that guarantees a sound business environment is properly applied. The relevance of the PCA has been fulfilled by adequately reducing the dimensionality and determining the direction of the data set (Cutler 2015; Dănescu and Prozan 2015). Management decision regarding this outcome would definitely take into account the entire data set made of five constructs seeking to explain the dependent variable regardless of the percentage variance assigned to each construct (Qiu et al. 2018). This suggests that management's motivation to tackle risk should include applying segregation of duties 20.078%, review 12.037% and information and communication 3.255% which constitute the least variance but are highly influential among detected control measures (Pellegrini et al. 2017).

**Table 2.** Principal component; total variance explained.

| Component | Initial Eigenvalues | | | Extraction Sums of Squared Loadings | | |
|---|---|---|---|---|---|---|
| | Total | % of Variance | Cumulative % | Total | % of Variance | Cumulative % |
| 1 | 1.804 | 36.088 | 36.088 | 1.804 | 36.088 | 36.088 |
| 2 | 1.427 | 28.543 | 64.631 | 1.427 | 28.543 | 64.631 |
| 3 | 1.004 | 20.078 | 84.709 | 1.004 | 20.078 | 84.709 |
| 4 | 0.602 | 12.037 | 96.745 | | | |
| 5 | 0.163 | 3.255 | 100.000 | | | |

Commitment to ethical standards is a preventive approach to managing risk and possible internal control challenges which has the most significant eigenvalue (36.088%) out of total variance of 84.709%. This is an indication that public sector requires reinforcement of ethical standards particularly the code of conduct, public sector Administration Acts, the constitution and other legislative instruments supporting the establishment of each public sector organization (Demek et al. 2018). Control environment explained 28.54% out of the total variance, and it is significant that control environment forms the foundation of internal control and the atmosphere on which other elements of control are built. However, a 28.54% variance by control environment is a remarkable indication that the absence of risk management could render the organization vulnerable to risk (Asiedu and Deffor 2017). The result emphasize that setting the right tone for the organization such as influencing risk appetite or management attitude towards risks and ethical values should be the priority (Arena et al. 2017; Asiedu and Deffor 2017). Based on the predictive power of the PCA, management can develop risk management polices around these three principal components to ensure that the public sector is fully conscious and embedded with the fundamental guidelines proposed by the COSO integrated framework, though not mandatory, is very influential.

4.1.2. Discriminant Analysis

In order to confirm the outcome of the result, we performed a discriminant analysis based on the same predicting ability as the PCA, to measure the direction of the data, if put in groups to separately explain the dependent variable, as was done in a northern city of China by (Dong et al. 2017). Nevertheless, the pseudo R-square (Table 3), shows a highly significant relationship between the explained variable and the independent factors at 91% variance based on the Nagelkerke's $R^2$ (Aziz et al. 2015; Balabonienė and Večerskienė 2015).

**Table 3.** Pseudo R-square test of discriminant groups, using the Nagelkerke function.

| | |
|---|---|
| Cox and Snell | 0.940 |
| Nagelkerke | 0.906 |
| McFadden | 0.916 |

Link function: Logit.

The Nagelkerke's test score explains the level of relationship at a proportion variance of 91%, when we consider the regressors as one large determinant group (Delaplace et al. 2015). Nonetheless, the emphasis is on the total variation explained by the model $R^2$ as the basis to inform management decisions in relation to its strategic objectives, operational objectives, financial reporting and compliance objectives, bearing in mind the impact of these three components on enterprise risks. Meanwhile, management must make provision for risk when planning, organizing, leading, coordinating and controlling the organization's activities in order to minimize the effects of risks on capital and earnings based on the direction of the model and individual variables (Boe and Kvalvika 2015; Bromiley et al. 2015). The pseudo R-square computes a good $R^2$ value which could facilitate management to plot a path and then develop tools and techniques to remain on the path using the predicting characteristics of the model (Paino et al. 2015). On the other hand, the chi-square test (Table 4), illustrates a strong possibility to reject the null hypothesis at a highly significant level. Going by this outcome, we may reject the null hypothesis at 0.000 *p*-values, taking into account the model reliability and the goodness of fit, the direction of the data set and the degree of association. The correlation coefficient between the dependent and independent variables are valuable points to note regarding management's policies about risk (Boe and Kvalvika 2015).

**Table 4.** Chi-square; test of parallel lines goodness of fit.

| Model | −2 Log Likelihood | Chi-Square | Df | Sig. |
|---|---|---|---|---|
| Null Hypothesis | 0.000 | | | |
| General | 0.000b | 0.000 | 1472 | 1.000 |

Consequently, the Walk's lambda test of variables (Table 5), thus ($C = C_{k0} + C_{K1}X_1 + C_{K2}X_2 + C_{K3}X_3$ ...), illustrates total contribution attributed to the constructs in accordance with the discriminant function (Bosten et al. 2017). The lambda test of F-statistics infers the statistical significance of the model (Ahsan and Rahman 2017). The test proves significant as the corresponding *p*-values to each variable are less than 0.05 (Grover et al. 2018). Based on this result we could precisely conclude that the explanatory variables being control environment, commitment to ethics, review and information and communication accurately explained a positive relationship with the dependent variable. However, the test result also represents a linear function per the assumptions underlining the model (Grover et al. 2018). Nevertheless, it is a summary of determinants of effective risk management elements, describing the likelihood of improvement in the performance of public sector entities at a significant level of 0.000. It also suggests that, any strong principal assumption enforcing management desire to seek a perfect risk management approach must be based on the COSO integrated framework.

**Table 5.** Wilks' lambda.

| Number of Variables | Lambda | df1 | df2 | df3 | Exact F | | | | Approximate F | | | |
|---|---|---|---|---|---|---|---|---|---|---|---|---|
| | | | | | Statistic | df1 | df2 | Sig. | Statistic | df1 | df2 | Sig. |
| 1 | 0.331 | 1 | 17 | 281 | 33.410 | 17 | 281.000 | 0.000 | | | | 0.000 |
| 2 | 0.169 | 2 | 17 | 281 | 23.586 | 34 | 560.000 | 0.000 | | | | 0.000 |
| 3 | 0.135 | 3 | 17 | 281 | | | | 0.000 | 15.647 | 51 | 831.433 | 0.000 |

The rejection of the null hypothesis inferring the prominence of the significant coefficients.

Again, the result also suggest that, the outcome of risk management could probably lead to strengthening the internal control systems as well as aligning future policies to the nature of risk in the public sector (Lawson et al. 2017). Meanwhile, based on the choice of independent factors considered for the analysis, the test result in itself is no guarantee of any structural change unless management reforms its internal control design to cover the entire operation of the organization such as the board, committees, management, divisions and unit levels to function towards the direction of the strategic objectives and governance principles.

*4.2. Discussion and Theoretical Implication*

The outcome of this study has endorsed the call for structural reforms at various operational levels across the vast scope of the public sector. The value of sound public sector risk management and internal control systems comes with positive implications (Ahluwalia et al. 2016; Ahmed 2016; Akotia 2016). First, the result (Table 2) reaffirms the importance of demonstrating a commitment to ethical values and a control environment as key determinants of positive risk attitude that would align the activities within the organization to ethical practices that will minimize the occurrence of risk (Ashraf and Uddin 2016; Axelsen et al. 2017). The study also makes a relevant contribution to literature by creating a research gap for further debates regarding the relationship between internal control principles and risk management in the private sector and other jurisdictions, since it is not a popular topic in literature. Previous studies have barely discussed this relationship as many often refer to the concepts interchangeably (Earley et al. 2016; Elbanna 2016; Dong et al. 2017). Based on these revelations, the study also recommends extensive research to confirm the relationship between corporate governance and risk management as it was fairly discussed (Basu 2016; Baumgartner and Rauter 2017). According to the COSO internal control framework, it is the sole responsibility of the board and top management to establish the risk culture, control environment, and demonstrate commitment to these sets of values as a positive example for the entire organization and for the purposes of compliance and other corporate objectives (Adhikari and Gårseth-Nesbakk 2016; Agyei-Mensah and Kwame 2016). This certainly constitutes a potential topic for further research. In a situation where the corporate governance principles do not support risk management and an unfortunate event happens, the question arises as to who bears the losses in the case of the public sector where losses cannot be allocated to individuals working in the capacity of public servants (Barafort et al. 2017). Drawing from the assumption that the public or the taxpayers are not directly involved and in contact with the day-to-day activities of the public organization, there is a need to ensure that directors or public servants work in the best interest of the public by applying certain acceptable principles necessary to create value for money (Anna and Nikolay 2015; Arundel et al. 2015). In a related discussion, Afiah and Azwari (2015) stated that this warrants the existence of corporate governance in every organization whether private or public to serve as a positive indication of trust between stakeholders, creditors and owners of an entity. The positive implication of this study will direct decisions of internal auditors, the board and management to view the existence of risk management as relevant tools and techniques to address the issues of corporate governance from the perspective of principal stakeholders whose interest is to be protected (Gantz and Philpott 2013). As the discussion involves the public sector, the principal stakeholders who are the taxpayers will stand to benefit if the public entities exercise sound internal control and risk management practices that will secure trust and confidence in public intuitions (Themsen and Skærbæk 2018). The study reassures stakeholders in the public sector that the existence of internal control elements such as the segregation of duties, review, commitment to ethics, control environment and communications and information are clear examples of good corporate governance indicators needed to be considered to minimize the impact of risk and enhance performance and effectiveness (Asiedu and Deffor 2017). Looking forward, the study also suggests that the poor performance of public organizations, particularly in Ghana, should not always be measured based on macroeconomic indicators, according to some arguments

by Domingues et al. (2017), and sustainability could also be based on certain internal factors such as control, good governance and risk management practices.

## 5. Conclusions

According to the custodians of internal control systems, the Committee of Sponsoring Organizations of the Treadway Commission—COSO, an entity's engagement in controlling, prevention and detecting risk is a management activity designed to guarantee reasonable assurance regarding operations and certainty that the objectives will be achieved (King 2016). Based on this concept, we hypothesized the impact of risk on the performance of public entities in Ghana, and to establish a statistical linkage to fully integrate internal control systems with risk management. This research is one the few studies in this discipline which make a principal contribution suggesting that internal control elements are good determinants of effective risk management in the public sector at a highly significant level. Similar studies in the UK by Daniela and Attila (2013) have made good contributions suggesting that the audit committee and the board could enforce risk management, but Lawson et al. (2017), believed this could only happen in advance economies where public sector employees are already embedded with risk and internal control practices. Even in countries where internal control is popular, it requires the effort of specialized institutions to perform oversight duties by monitoring and supervising (Pellegrini et al. 2017). Although a control environment which happened to be the responsibility of the board explained 28.54% of variance, this is not sufficient based on the findings by Appiah et al. (2016), who hold a separate view stating that, the board is autonomous and does not directly influence the day-to-day activities of the organization, which implies that based on our result even if a control environment explains the state of internal control, it is no guarantee that there is 100% assurance of the absence of control deficiencies. A second opinion also stated that the board lacks the capacity and time to supervise and fully implement risk policies across the entire organizational setup through management to the rank and file. Callahan and Soileau (2017), therefore, discredited the role of the board, citing that the members are external parties to the organization and secondly, the number of times board meeting are scheduled to deliberate on issues bothering on the future of the organization is not sufficiently adequate to perform a thorough assessment such as internal control and risk management issues which demand a holistic approach. Considering the above issues and judgment from separate opinions, it is therefore recommended that management and policy makers take decisions in view of the five explanatory factors and base the reforms on total variance rather than selectively approaching risk by isolating the variables. The public sector is very vast and risk management can be challenging in the midst of increasing government projects, globalization and the consequence of inter-governmental transactions through the public sector (Woods et al. 2017). The scope of responsibilities of auditors and those in charge of governance has increased tremendously in recent times, making it practically impossible to validate each activity that is performed in order to identify potential risk (Da Silva Etges et al. 2018). Looking forward, we believe that an auditor's role in risk assessment should be supplemented with computerized internal control systems, segregation of duties, periodic review, and communication of timely information within the public sector in order to minimize the occurrence of risk and the impact on performance. Although the COSO framework is not mandatory, it is highly influential and many have enjoyed its benefits.

**Author Contributions:** The main supervisory work was done by Y.K. (Professor); P.Y.L. performed literature review and data analysis while F.B.M.B. supported with questionnaire analysis. Review of the final draft and editing was done by N.B.B. (Professor).

**Funding:** The research is fully supported by the National Natural Science Foundation of China, under the grant 71371087 and Jiangsu University.

**Conflicts of Interest:** We wish to declare no conflict of interest with regards to this work.

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
