# Peer review of "The Value of Public Sector Risk Management: An Empirical Assessment of Ghana"

_admsci, doi:10.3390/admsci8030040_

Round 1
Reviewer 1 Report
Dear authors,
thank you for letting me read this paper which deals with a very relevant topic.
The paper is well written and clear. A final proof-reading is necessary to check for some errors, typos and other minor editing and formatting problems.
Piori to publication, I suggest the following revisions:
1) the introduction does not make fully clear what is the aim and/or the research question to be investigate in particular.
2) Methodology. The hypothesis appears out of the blue. They are not clearly linked to any literature. I suggest to move the hypotesis in the previous section, by linking the literature review with the hypotesis, that is to explain better the supporting literature. Besides, the hypothesis as they stand are not fully clear, for instance what do the authors mean for "positive impact on risk"? The dependent variable "Effectivenness" is not explained hot it is measured, so are the other variables. I suggest to attach as an appendix the questionnaire used.
3) the results need a discussion section. What the results add the literature? What are the theoretical and practical implications of the findings? Please integrate.
Some minor issues:
-check for the references in the text, for example you cite (Sterck M, 2005), but in the reference there is Sterck, M, BS, G. Bouckaert (2005). Please check for consistency and pay attention to the order "Name, Surname".
- The figures need to be numbered. The first oen "COSO..." is not clear what does it add to the text. I suggest to remove it, if don't engage with it.
- When you refer for the first time to the COSO you should write in full, then use COSO consistently. For instance, you write in full at page 5, after using COSO many times before.
- The extract from Knechel and Salterio (2016) should have the page number.
Author Response
Response to Reviewers Feedback
We appreciate your feedback regarding our paper. Our attention has been fully drawn to a few critical issues under different sections that needed to be addressed. The major revision will be effected in the revised manuscript. In the meantime, here are the responses to the editor’s remarks per the guidelines.
1. The introduction is not very clear on the research question/aim to be investigate in particular.
In the build up to this study we considered a wide range of issues bothering the performance of the public sector as against their counterparts in the private sector, before finally settling on risk and the fact that the public sector of Ghana is in urgent need for reinforcement of internal control and risk management. In the wisdom of the authors, any form of reforms to salvage the public sector of would require a solid foundation build on ethical standards and commitment to a structural approach to identifying risk from the top to bottom assessment. In this regard we aim to get information from management and employees acting in various capacities to give us a feedback by means of self-assessment through by answering our questions. This gave us the sound foundation for exploring the risk management processes in the public sector and the role of internal control in ensuring effective risk management. We also believe that internal control is more developed and popular than risk management and since the COSO framework has made it known that risk management should always form an integral part of internal control, we applied both concepts to achieve efficiency. We also bear in mind that internal control focuses on compliance, while risk management focuses on performance efficiency, therefore we assumed a research question seeking to investigate the impact of internal control on organisational risk management practices. We made use internal control elements as the independent variables which are traditionally not described as risk management elements but very dominant in advocating risk effective risk management.
In evaluating the various aspects of organizational functions that could constitute risk, the study aimed at drawing the attention of government , stakeholders, internal auditors , public sector workers , policy makers and the academic community about the need to establish a risk management program in the public sector . Although from a small beginning, the approach would complement the existing laws and constitutional provisions such as the legislative instruments, public administrative Acts and the financial administrative Acts that provides clear guidelines to the running of public organisations in Ghana. In spite of Ghana having one of the most attractive legal and constitutional environment capable of guiding the conduct of business and ensure compliance, the public sector is distressed with gross disregard for internal control and its application. A major obstacle to reforms and proposals put before government by various stakeholders including some international creditors such as the IMF and the World failed to materialise and it’s partly due to deficiencies associated in the internal control structures which spells the rules for risk management. Ghana can boast of having the most beautiful legal provisions in sub-Sahara Africa that guides the public sector to be more efficient in terms of good governance and guarantee value for money for the taxpayers. However, due to weak institutions and lack of control enforcement, the public is left at the mercy of technocrats in charge of affairs at public entities.
With this background, the authors sought to propose a more robust but simple top to down self-assessment in a structural approach that will identify the existence of internal control and risks management in the public sector and ensure that the concept is well grounded and fully functional in the public sector.
2. Methodology. Hypothesis are not clearly linked to literature.
The hypothesis were deduced from a wide range of literature relating to internal control and risk management but closely linked to the conceptual framework and the mind map we developed based on literature (see below). As we illustrate in the conceptual framework below, the study did not address risk management in isolation from internal control itself, hence the decision to draw literature which extensively discussed both concepts, which is why we subsequently could not direct associate the hypothesis to a particular literature. The sub headings form an umbrella term for both concepts which we believe has accommodated the variables and the hypothesis in general without crediting each of them to a particular section of the literature. We deployed what we termed as a combination of traditional and non-traditional elements risk management. Each hypothesis is related to either a non-traditional element or those performing a purely risk management role per the COSO framework and the vast literature covered. The decision to combine the two disciplines was based on the fact that risk management forms an integral part of internal control. In this regard we are guided by introducing a reform and innovation that will at the end of the day give us a fair evaluation of the existence of control structures in the public sector and the important roles they play in creating the platform to promote risk awareness. However, the linkage between the hypothesis and the literature review will be looked at for proper alignment.
A typical design of management functions depicting internal control.
The Mind Map Guiding the literature
Proposed by Authors
Guided by the mind map , there are various control and risk elements that could form the basis of our hypothesis . Take for example , segregation of duties which happens to be a purely a control activity meant to address internal control lapses is seen performing a non traditional roles and its fairly discussed under risk response . Same applies to review and the rest of the variables which were used interchageably in the discussion . Haven said that , we point raised by the reviewers will be dealth with in the revised manuscript .
3. Discussion section , contribution to literature and theoritical implications
This is obviously an important point raised by the reviewers which must not be taken for granted. This will form part of the revised version. The findings implications and contribution to existing knowledge will all be dealt with in the revised manuscript. However, evidently On the other hand, it was shown that clearly developed and recognized internal control structures could stimulate risk identification, evaluation, management and monitoring, as well as the prevention and detection within an organisation.
Minor Issues
All the points are well noted and will be addressed appropriately. The references and the inappropriate use of COSO, will equally be addressed. The methodology will also be given a proper clarity.
v
